# The Challenge of Bacteremia Treatment due to Non-Fermenting Gram-Negative Bacteria

**DOI:** 10.3390/microorganisms11040899

**Published:** 2023-03-30

**Authors:** Svetlana Sadyrbaeva-Dolgova, María del Mar Sánchez-Suárez, Juan Antonio Reguera Márquez, Carmen Hidalgo-Tenorio

**Affiliations:** 1Pharmacy Service, Hospital Universitario Virgen de las Nieves, 18014 Granada, Spain; 2Instituto de Investigación Biosanitaria ibs.GRANADA, 18014 Granada, Spain; 3Microbiology Service, Hospital Universitario Virgen de las Nieves, 18014 Granada, Spain; 4Infectious Disease Service, Hospital Universitario Virgen de las Nieves, 18014 Granada, Spain

**Keywords:** multidrug resistant non-fermenting gram-negative bacteria, appropriate empirical treatment, nosocomial bloodstream infections

## Abstract

Nosocomial infections caused by non-fermenting Gram-negative bacteria are a real challenge for clinicians, especially concerning the accuracy of empirical treatment. This study aimed to describe the clinical characteristic, empirical antibiotic therapy, accuracy of these prescriptions for appropriate coverage and risk factor for clinical failure of bloodstream infections due to non-fermenting Gram-negative bacilli. This retrospective, observational cohort study was conducted between January 2016 and June 2022. Data were collected from the hospital’s electronic record. The statistic tests corresponding to each objective were applied. A multivariate logistic regression was performed. Among the total 120 patients included in the study, the median age was 63.7 years, and 79.2% were men. Considering the appropriate empirical treatment rate by species, inappropriate treatment for *S. maltophilia* was 72.4% (*p* = 0.088), for *A. baumanii* 67.6% and 45.6% for *P. aeruginosa*. Clinical success was achieved in 53.3%, and overall, 28-day mortality was 45.8%. ICU admission, sepsis or shock septic, age, previous antibiotic treatment and contact with healthcare facilities were independently associated with clinical failure. In conclusion, bloodstream infection produced by multidrug-resistant non-fermenting Gram-negative bacteria is a significant therapeutic management challenge for clinicians. The accuracy of empirical treatment is low due to the fact that it is not recommended to cover these microorganisms empirically, especially *S. maltophilia* and *A. baumanii*.

## 1. Introduction

During the last decade, there has been an increased incidence of infections caused by multidrug-resistant Gram-negative bacteria, including multidrug-resistant Enterobacteriaceae, *Pseudomonas aeruginosa* and *Acinetobacter baumannii*. These microorganisms are generally implicated in severe infections and currently represent a major global public health problem [1]. Infections caused by these microorganisms have a worse prognosis than those caused by sensitive pathogens due to the fact that antimicrobial treatments initiated before microbiological data are available to determine or confirm the etiology of the process are ineffective in a significant number of cases and, in addition, the type of treated patient is often very complex [2].

The inappropriate or excessive use of broad-spectrum antimicrobials like carbapenems led to the emergent appearance of resistant bacteria to these antibiotics. Carbapenem resistance of Gram-negative bacteria became a global problem. To the point that in the global priority list of antibiotic-resistant bacteria published by the World Health Organization in 2017, three of the four microorganisms designated as critical priority for research and development of new antibiotics are carbapenem-resistant pathogens, including carbapenem-resistant *Enterobacteriaceae* or carbapenemase-producing *Enterobacteriaceae*, carbapenem-resistant *Pseudomonas aeruginosa* and *Acinetobacter baumannii*. These microorganisms are the most important carbapenem-resistant Gram-negative bacteria in clinical practice due to the increasing incidence of these bacteria worldwide in recent years, the lack of alternative antimicrobials for therapy, and the high mortality rates associated with these infections [3].

Increased antimicrobial resistance and limitations in the development of new antibiotics (especially against Gram-negative bacteria) lead to an increasing lack of therapeutic options for treating these infectious diseases. That requires using older antibiotics, as colistin, with a significantly increased toxicity rate [4,5,6].

Bloodstream infection (BSI) is a pathological entity associated with a high morbidity and mortality rate and represents an increasing public health concern. The prognosis and evolution of the infection depend directly on factors such as the patient’s baseline situation, the severity of its presentation, the causative pathogen and its early identification, and the implementation of correct empirical or targeted treatment [7,8]. The most severe presentation of BSI is septic shock, where the time until antibiotic treatment administration impacts patient survival [9,10,11]. BSI due to multidrug-resistant microorganisms occurs in patients previously colonized by these bacteria. These patients tend to be long-term, immunocompromised, patients admitted to the ICU and those undergoing invasive procedures [12,13].

Non-fermenting Gram-negative bacilli (NFGNB) are ubiquitous in nature, found in the soil or water, and as commensals on human skin or in the gut. In the hospital environment, they may be isolated from humidifiers, ventilator machines and accessories [14]. These microorganisms are frequent causes of nosocomial infections and represent a potential danger for severe patients with altered immune systems, such as subjects with cancer or admitted to the intensive care unit. These microorganisms represent several challenges for clinicians besides the limited therapeutic options available. Another is the difficulty differentiating colonization from infection in critically ill patients [15].

The selected regimen for nosocomial infections and some healthcare-associated infections should include an antibiotic with adequate activity against *Pseudomonas aeruginosa*. However, empirical treatment in critically ill patients against *Acinetobacter* spp. and *S. maltophilia.* is not recommended as it may delay appropriate treatment [15,16].

The increase in multidrug-resistant pathogens, in turn, implies the use of combined or broad-spectrum antibiotherapy, especially in nosocomial infections and in situations of severe sepsis or shock. However, to date, there is controversy about the use of monotherapy or combination therapy in the targeted treatment of these microorganisms [17,18,19,20,21]. Even so, the treatment regimens for these microorganisms are based on combined treatments with older drugs such as Colistin, Fosfomycin or Tigecycline. Fortunately, new antibiotics have appeared recently, such as Ceftolozane-Tazobactam, Ceftazidime-Avibactam and Cefiderocol, with better clinical results due to their spectrum and PK/PD characteristics.

This study aimed to describe the clinical characteristic, empirical antibiotic therapy, accuracy of these prescriptions for appropriate coverage and risk factor for clinical failure of bloodstream infections due to non-fermenting Gram-negative bacilli.

## 2. Materials and Methods

### 2.1. Study Design and Population

This retrospective, observational cohort study was conducted in a 1000-bed tertiary-level university hospital in southern Spain between January 2016 and June 2022.

The study included adult patients (≥18 years of age) with bacteremia caused by non-fermenting Gram-negative bacteria such as multidrug-resistant *Pseudomonas aeruginosa*, extensively-resistant (XDR) *Acinetobacter baumannii* and *Stenotrophomonas maltophilia*.

Multidrug resistance for *P. aeruginosa* was defined as nonsusceptibility to at least one antibiotic in at least three classes for which *P. aeruginosa* susceptibility is generally expected [22]. Extensively drug-resistant *A. baumannii* was defined as susceptibility limited to at least two categories [23]. All isolates with *S. maltophilia* were considered difficult to treat due to intrinsic resistance mechanisms to many categories of antimicrobials [15].

Data were included only for microorganisms with available sensitivity data.

In cases with multiple bacteremic episodes, only the first episode was included for each patient.

Patient data were collected and processed according to the current regulations of Organic Law 7/2021, of 26 May, on the protection of personal data processed for the purposes of prevention, detection, investigation and prosecution of criminal offenses and the execution of criminal sanctions. The study was approved by the local ethics committee in compliance with the Declaration of Helsinki regulations and biomedical research legislation (Law 14/2007, 3 July).

### 2.2. Variables and Definitions

Data were collected from the hospital’s electronic record, containing episode-related demographic, microbiological and prescription data. The data was: age and sex, comorbidities and baseline disease, source of bacteremia, causative agents and their antibiotic susceptibility, antibiotic treatment, severity at onset, mechanical ventilation requirement, and 14- and 28-day mortality.

All infectious syndromes included were diagnosed based on clinical, analytical and imaging data. The comorbidity burden was assessed using the age-adjusted Charlson comorbidity index. The severity of BSI was assessed using Pitt score. Nosocomial bacteremia was defined as those detected 48 h or more after admission to the hospital. Healthcare-associated bacteremia was defined when the patient has one of the following criteria: recent hospitalization, admission from a long-term-care facility, and undergoing chronic hemodialysis or intravenous treatment.

Previous colonization by these multidrug-resistant microorganisms (MDRO) was defined as the detection of rectal or respiratory colonization in surveillance cultures in the 90 days before the bacteremia episode.

Prior antibiotic therapy was defined as the usage of any antimicrobial agent for more than two days. Time to appropriate antibiotic therapy was measured by the period from blood culture extraction and the appropriate antibiotic agent prescribed.

The treatment was considered appropriate if the microorganism was susceptible to the antimicrobial agent *in vitro*; an antimicrobial treatment delay of more than 24 h from blood culture extraction was considered inappropriate.

Combination therapy was considered two or more agents active against Gram-negative microorganisms.

The primary outcome variable was clinical success, defined as survival and absence of recurrence at day 14 following the onset of bacteremia, resolution of signs and symptoms of infection. Secondary outcomes included 14 and 28-day mortality, re-admission within 30 days after discharge, duration of the treatment, hospital stay, and the onset of *C. difficile* infection during hospital stay.

### 2.3. Antibiotic Susceptibility

Antimicrobial susceptibility of microbiological isolates was determined with a microdilution system (MicroScan walkAway. Beckman Coulter) or the E-test method (Liofilchem^®^ MIC Test Strip) according to the current breakpoints recommended by the Clinical and Laboratory Standards Institute (CLSI) till 2020 and after by the European committee on antimicrobial susceptibility testing (EUCAST). The carbapenemase resistance detection was performed by qualitative lateral flow immunoassay (NG-Test^®^ CARBA-5).

Surveillance for *C. difficile* infection was performed during the patient’s hospital stay, and this infection was diagnosed by toxin A/B rapid immunoassay.

### 2.4. Statistical Analysis

In the descriptive analysis, the mean and standard deviation, or median and interquartile range, were calculated for quantitative variables, and absolute and relative frequencies were calculated for qualitative variables. The Pearson Chi-square test or Fisher’s exact test was used to analyze relationships among qualitative variables. The non-parametric Mann–Whitney test was used to examine relationships between qualitative and quantitative variables with a non-normal distribution (assessed by Kolmogorov–Smirnov test). A multivariate logistic regression model was constructed by backward stepwise selection, considering entry criteria of *p* ≤ 0.05 and exit criteria of *p* > 0.1. It included all variables that were statistically significant in bivariate analysis or otherwise considered relevant. IBM SPSS Statistics version 19 software program was used for the statistical analysis; *p* ≤ 0.05 was considered significant in all tests.

## 3. Results

### 3.1. Baseline Characteristics of the Patients

Among the total 120 patients included in the study, the median age was 63.7 years, and 79.2% were men. The median of the Charlson index was 4 (IQR 3–6). In terms of the patients’ underlying diseases, most of them were cancer patients (35.8%), immunosuppressed (31.7%), with diabetes (24.2%) and with respiratory pathologies (18.3%), among others.

The most frequent source of infection was respiratory (36.7%) followed by catheter-associated bacteremia (14.2%) and urinary foci (14.2%). Patients with severity criteria were 40.8% with sepsis or septic shock and with Pitt score > 4 (31.7%). Almost all patients underwent an invasive procedure (94.2%).

Most patients (69.2%) with these isolates were admitted to medical or surgical wards. More than half (61.7%) of the patients received antibiotic treatment within 30 days prior to admission. 36.7% were hospitalized within the last 30 days and 31.7% had contact with healthcare system. The median time of onset of bacteremia since admission was 22 days. Regarding antibiotic treatment, 86.7% of the patients were receiving antibiotic treatment at the onset of bacteremia. 62.5% of treatment for bacteremia was monotherapy. The rate of appropriate treatment at the time of the onset of bacteremia was 42.5%. The median of duration of antibiotic therapy by bacteremia was 11 days (IQR 4–17) and the median of length of hospital stay per patient was 40.5 days (IQR 23.0–68.8). The rest of the baseline variables are shown in Table 1.

### 3.2. Bacterial Isolates and Antibiotic Sensitivity

The most frequently isolated microorganism was *P. aeruginosa* (47.5%), followed by *Acinetobacter* spp. (28.3%) and *S. maltophilia* (24.2). 14.2% of patients were previously colonized by these microorganisms (Table 1).

98.2% of *P. aeruginosa* isolates were resistant to Imipenem and 70.2% to Meropenem, 31.5% to Piperacillin-Tazobactam and 43.9% to Cefepime, among other anti-pseudomonic alternatives. Although only 22% were carbapenemase producers, mainly were metallo-beta-lactamase producers (14.0% were IMP-type carbapenemase producers, and 7.0% were VIM-type producers).

Among *A. baumannii* isolates (28.3% of total), 38.2% were OXA-23-type carbapenemase producers. 5.9% of *A. baumannii* isolates were resistant to Colistin and 94.1% to Amikacin.

*S. maltophilia* isolates were resistant to Levofloxacin (27.6%) and Trimethoprim-Sulfamethoxazole (6.9%). The sensitivity to antibiotics is shown in Table 2.

### 3.3. Antibiotic Treatment

Regarding the prescribed antibiotherapy: the most prescribed antibiotics for bacteremia produced by *P. aeruginosa* were Piperacillin/Tazobactam (19.56%), Amikacin (14.7%), Ciprofloxacin (11.96%) and Meropenem (11.4%). For *A. baumanii* bacteremia, the most frequently used antibiotics were Meropenem (26.8%), followed by Colistin (25.7%) and Tigecycline (17.5%). In case of *S. maltophilia* bacteremia, the most prescribed antibiotics were Trimethoprim/Sulfomethoxazole (21.1%) and Levofloxacin (17.8%), followed by Meropenem (15.6%) and Tigecycline (11.1%). In recent years, new antimicrobials have been released on the market. Of these, Cefiderocol was prescribed for four patients with BSI produced by *A. baumannii*; Ceftolozane-Tazobactam was prescribed for ten patients with BSI caused by *P. aeruginosa* and one patient with BSI caused by *S. maltophilia*. Ceftazidime-Avibactam was prescribed for twelve patients. Seven of these prescriptions were for BSI caused by *P. aeruginosa*, three for BSI caused by *S. maltophilia* and two for BSI caused by *A. baumannii*. Two of the total Ceftazidime-Avibactam prescriptions were associated with Aztreonam. 

Considering the accuracy of empirical treatment rate by species, appropriate treatment was 62.0% for *P. aeruginosa,* 22.0% for *A. baumanii* and 16% for *S. maltophilia* (Table 3).

### 3.4. Variables Associated with Clinical Success

Clinical success was achieved in 53.3%, and overall, 28-day mortality was 45.8%.

The clinical success rate was lower for ICU patients (14.1% vs. 85.9%, *p* < 0.001), with signs of sepsis or shock septic (23.4%, *p* < 0.001) or with Pitt score ≥ 4 (18.8%, *p* = 0.002). Also, it differs for patients receiving monotherapy or combination therapy (71.9% vs. 28.1%, respectively, *p* = 0.037). Moreover, the success rate was lower in patients who received antibiotherapy within the previous 30 days (50% vs. 75.0%, *p* = 0.008) and patients in contact with healthcare facilities (23.4% vs. 41.1%, *p* = 0.049) (Table 4).

Duration of antibiotic therapy for BSI (11 days longer, *p* < 0.001) and length of hospital stay were more prolonged in patients successfully treated (21 days longer, *p* < 0.001). Re-admission rate was also higher in this group of patients (25% vs. 5.4%, *p* = 0.005). No difference was found in terms of appropriate empirical treatment (*p* = 0.854). The overall incidence of *C. difficile* infections was 4.2%, and in patients with clinical success 3.1% vs. 5.4% with clinical failure (*p* = 0.663). (See Table 4).

### 3.5. Risk Factors Associated with Clinical Failure

Multivariate analysis was performed to identify the independent factors associated with clinical failure. ICU admission (OR 9.61, 95%CI 2.89–31.95, *p* < 0.001), sepsis or shock septic were associated with increased clinical failure (OR 5.19, 95%CI 1.80–14.96, *p* = 0.002). Others factors such as age (OR 0.952, 95%CI 0.919–0.987, *p* = 0.007), previous antibiotic treatment (OR 6.04, 95%CI 1.92–19.02, *p* = 0.002) and contact with healthcare facilities (OR 3.157, 95%CI 1.12–8.88, *p* = 0.029) were independently associated with clinical failure (Table 5).

## 4. Discussion

Non-fermenting Gram-negative microorganisms are common in the hospital environment and represent a danger for patients with immunodeficiencies (oncologic or hematologic patients) or critically ill intubated patients. The treatment of bacteremia caused by carbapenem-resistant NFGNB represents a major challenge for clinicians. Clinical success was observed mostly in non-critical patients admitted to medical-surgical facilities.

Although these microorganisms have limited therapeutic options, in our study, we found that there are factors that may contribute to clinical failure, like age, critical condition of the patient, recent antibiotic treatment or contact with the health care system. In line with data obtained in other studies, less severe patients had a better clinical success rate [24].

All-causes in-hospital mortality in our study was 56.7%, and 28-day mortality was 45.8%. These data are similar to the data already published [25,26]. The above is due to the complexity of these patients, prolonged hospitalization, ICU admissions, and baseline situation prior to bacteremia produced by these microorganisms. Furthermore, in line with our study results, the most common foci of bacteremia were respiratory and catheter-associated. In addition, factors that were independently associated with mortality, such as old age and high Pitt score. Also, the association between inappropriate empirical treatment and clinical failure was not found [26].

Regarding empirical treatment, in our study, less than half had appropriate treatment. However, the overall accuracy rate is relatively high considering the extremely demanding sensitivity profiles of isolates microorganisms and compared with previously published data [16,27,28]. In one of the studies [16], the rate of inappropriate empirical first-line treatment of bacteremia caused by difficult-to-treat Gram-negative bacteria, including *A. baumannii*, was very high and directly related to mortality. This is probably due to the fact that before presenting BSI, the patient already had isolations from other foci with the same microorganisms or as colonizers in epidemiological surveillance cultures before producing an infection. Data support that in patients colonized by multidrug-resistant bacteria, factors such as admission to the ICU, treatment with chemo- or radiotherapy, and invasive abdominal procedures can produce the onset of bloodstream infections by these same microorganisms [13]. 14.2% were previously colonized in our data, which could contribute to the prompt initiation of adequate empirical treatment. Due to this reason and probably because of the small sample size, our study did not observe a statistically significant relationship between appropriate treatment and clinical success. However, this data observed in our study is in accordance with the results of previously published studies, the accuracy rate of appropriate empirical treatment decreases according to the resistance profile of the microorganism [8,29].

In our study, the percentage of *P. aeruginosa* isolates resistant to carbapenem was very high (98.2% for imipenem and 70.2% for meropenem), probably due to the loss of porins and expulsion pumps, as carbapenemase producers accounted for only 22% of the total. Therefore, the accuracy of empirical treatment was more than half of the patients (62.0%). According to the published recommendations [3,22], in patients with severe pathologies, immunodeficient, with multiple exposures to antimicrobials, especially carbapenems, in addition to the virulence mechanisms and ability to acquire resistance mechanisms of *P. aeruginosa* [30], to design an appropriate empirical regimen, with antibiotics such as Ceftolozane-Tazobactam or Ceftazidime-Avibactam. However, for *A. baumannii* and *S. maltophilia,* the accuracy rate was lower −22.0% and 16.0%, respectively- probably due to the absence of recommendations in many cases, especially in non-critical patients, to cover these patients empirically. On the other hand, to differentiate in a critical patient the infection from colonization, which could be the leading cause of delay and, therefore, inappropriate treatment.

Furthermore, our study also found that the most significant clinical success was in patients treated in monotherapy. This is probably because they were less severe patients. Therefore, clinicians generally avoid prescribing two antibiotics with Gram-negative activity in patients without signs of sepsis or septic shock. In addition, there is insufficient evidence on the benefit of combined treatment for infections produced by these microorganisms [5,19,20]. However, in severe patients with bacteremia due to Gram-negative bacilli producing carbapenemases, there is evidence that two active antibiotics are better than one [31]. In a recently published study aimed to compare different alternatives for the treatment of *S. maltophilia* in monotherapy, the clinical failure rate was similar between trimethorpim-sulfamethoxazole (TMP/SXX), minocycline or fluoroquinolones. Indeed, the clinical failure rate was slightly higher with TMP/SMX and in a multivariate analysis, mortality was lower in minocycline treatment compared to TMP/SMX (OR 0.2, 95%CI 0.1–0.7) [25]. In our study, the most commonly used antibiotic was TMP/SMX, although we did not compare the treatment groups and possible implication of each in clinical failure. Therefore, these data led to reconsidering the use of monotherapy in severe patients infected by this microorganism and support to prescribe combination treatment with TMP/SMX and minocycline as already recommended [15].

A multicenter study performed by Kadri et al. with a cohort of 21,608 patients with BSI [8] assessed the accuracy of the empirical treatment, and the rate of discordant empirical antibiotic therapy was 19%. Although, for NFGNB, the discordant empirical treatment was higher: 83% for *S. maltophilia* and around 50% for *P. aeruginosa* and *A. baumanii.* These results are in line with the findings of our study, where the rates of inappropriate empirical treatment were 72.4%, 45.6% and 32.4%, respectively.

In our study, the length of hospital stay and duration of antibiotic treatment were longer in patients with clinical success. This data may be directly related to the survival rate in these patients, prolonged recovery due to complications that occurred during the hospital stay and longer treatment courses due to the underlying nature of these infections [24].

Our study has several limitations. Only blood isolates were considered without considering microorganisms isolated in other clinical samples from the primary focus of infection or microorganisms isolated in colonization. Another limitation of this study is its retrospective design in a single center. Therefore, it is challenging to extrapolate since each hospital manages these infections in a particular way, given that guidelines do not recommend the empirical use of therapy against these microorganisms.

However, the data from our study may provide evidence for managing infections caused by these microorganisms.

## 5. Conclusions

In conclusion, bloodstream infection produced by multidrug-resistant non-fermenting Gram-negative bacteria is a significant therapeutic management challenge for clinicians. Even using broad-spectrum antibiotics in empirical treatment, only half of the patients present clinical success with high mortality rate. The accuracy of empirical treatment is low due to the fact that it is not recommended to cover these microorganisms empirically, especially *S. maltophilia* and *A. baumanii*. Therefore, it is extremely important to consider the patient’s baseline condition, previous antibiotic treatment and contacts with the health care system in order to design the empirical treatment regimen.

## Figures and Tables

**Table 1 microorganisms-11-00899-t001:** Baseline characteristic of patients with bacteremia produced by NFGNB.

Characteristics of Patients	Total (n = 120)
Sex, male, n (%)	95 (79.2)
Age, years, mean (SD)	63.7 (56.0–74.3)
Age adjusted Charlson index score, median (IQR)	4 (3–6)
Setting: ICU Non-ICU	37 (30.8) 83 (69.2)
Comorbidities, n (%) Cancer Diabetes Liver disease Lung disease Neurological disease Immunodeficiency Gastrointestinal disease Hemodyalysis Surgery in 3 prior month	43 (35.8) 29 (24.2) 9 (7.5) 22 (18.3) 16 (13.3) 38 (31.7) 3 (2.5) 4 (3.3) 11 (9.2)
Sepsis/Shock septic	49 (40.8)
Invasive procedure, n (%)	113 (94.2)
Pitt score, median (IQR) Pitt scor ≥ 4, n (%)	2 (1–4) 38 (31.7)
Source, n (%) Respiratory Catheter-related Urinary Intraabdominal Skin &soft tissues Unknown foci Others	44 (36.7) 17 (14.2) 17 (14.2) 13 (10.8) 3 (2.5) 22 (18.3) 4 (3.4)
Previous MDRO colonization, n (%)	17 (14.2)
Admission in previous 30 days, n (%)	44 (36.7)
Antibiotic therapy in previous 30 days, n (%)	74 (61.7)
Contact with Healthcare facility, n (%)	38 (31.7)
Antibiotic therapy at onset of bacteremia, n (%)	104 (86.7)
Onset of bacteremia from admission, days, median (IQR)	22 (10–38)
Microorganisms, n (%) *P. aeruginosa* *A. baumannii* *S. maltophilia*	57 (47.5) 34 (28.3) 29 (24.2)
Combination therapy, n (%) Monotherapy	45 (37.5) 75 (62.5)
Appropriate treatment, n (%)	51 (42.5)
Clinical success, n (%)	64 (53.3)
Overall in-hospital mortality, n (%)	68 (56.7)
14-day mortality, n (%)	43 (35.8)
28-day mortality, n (%)	55 (45.8)
Duration of antibiotic therapy/bacteremia, days, median (IQR)	11 (4–17)
Length of hospital stay, days, median (IQR)	40.5 (23.0–68.8)
Incidence of *C. difficile* infection, n (%)	5 (4.2)
90-day re-admission, n (%)	19 (15.8)

SD: standard deviation, IQR: interquartile range; MDRO: multidrug-resistant microorganisms; ICU: Intensive care unit.

**Table 2 microorganisms-11-00899-t002:** Antibiotic sensitivity of NFGNB (%).

Antibiotics	*P. aeruginosa*	*S. maltophilia*	*A. baumannii*
Piperacillin-Tazobactam	68.5	-	-
Meropenem	29.8	-	-
Imipenem	1.8	-	-
Ciprofloxacin	49.2	-	-
Levofloxacin	33.3	72.4	-
Gentamicin	66.6	-	11.7
Amikacin	75.5	-	5.9
Tobramycin	75.4	-	11.8
Cefepime	56.1	-	-
Ceftazidime	63.2	-	-
Aztreonam	61.4	-	-
Colistin	70.2	-	94.1
Trimethoprim-Sulfamethoxazole	-	93.1	-

**Table 3 microorganisms-11-00899-t003:** Appropriateness of the treatment according to the isolated microorganism.

Microorganisms	Appropriate Treatment (n = 50)	Inappropriate Treatment (n = 70)
*P. aeruginosa*	31 (62.0)	26 (37.1)
*A. baumannii*	11 (22.0)	23 (32.9)
*S. maltophilia*	8 (16.0)	21 (30.0)

**Table 4 microorganisms-11-00899-t004:** Bivariate analysis of factors associated with primary outcome.

Variables	Clinical Success (n = 64)	Clinical Failure (n = 56)	*p*-Value
Sex, male, n (%)	53 (82.8)	42 (75.0)	0.369
Age, years, mean (SD)	62.64 (51.62–72.69)	64.73 (58.13–77.25)	0.157
Age adjusted Charlson index score, median (IQR)	4 (2–6)	4 (3–6)	0.076
Setting, n (%): ICU others	9 (14.1) 55 (85.9)	28 (50.0) 28 (50.0)	<0.001
Comorbidities, n (%) Cancer Diabetes Liver disease Lung disease Neurological disease Immunodeficiency Gastrointestinal disease Hemodyalysis Surgery in 3 prior month	19 (29.7) 19 (29.7) 3 (4.7) 12 (18.8) 9 (14.1) 17 (26.6) 2 (3.1) 3 (4.7) 6 (9.4)	24 (42.9) 10 (17.9) 6 (10.7) 10 (17.9) 7 (12.5) 55.3 (37.5) 1 (1.8) 3 (1.8) 5 (8.9)	0.182 0.142 0.301 1.000 1.000 0.240 1.000 0.622 1.000
Sepsis/Shock septic	15 (23.4)	34 (60.7)	<0.001
Invasive procedure, n (%)	61 (95.3)	52 (92.9)	0.704
Pitt score, median (IQR) Pitt scor ≥ 4	2 (0.0–2.75) 12 (18.8)	3 (2–5) 26 (46.4)	<0.001 0.002
Source, n (%) Respiratory Catheter-related Urinary Intraabdominal Skin & soft tissues Unknown foci Others	20 (31.2) 11 (17.2) 11 (17.2) 9 (14.1) 2 (3.1) 8 (12.5) 3 (4.7)	24 (42.9) 6 (10.7) 6 (10.7) 4 (7.1) 1 (1.8) 14 (25.0) 2 (3.6)	0.275
Combination therapy, n (%) Monotherapy	18 (28.1) 46 (71.9)	27 (48.2) 29 (51.8)	0.037
Previous MDRO colonization (in 3 prior month)	11 (17.2)	6 (10.7)	0.432
Admission in previous 30 days, n (%)	18 (28.1)	26 (46.4)	0.057
Antibiotic therapy in previous 30 days, n (%)	32 (50.0)	42 (75.0)	0.008
Contact with Healthcare facility, n (%)	15 (23.4)	23 (41.1)	0.049
Antibiotic therapy at onset of bacteremia, n (%)	53 (82.8)	48 (85.7)	0.803
Onset of bacteremia from admission, days, median (IQR)	23.5 (10.25–39.75)	21 (9.25–38)	0.683
Microorganisms, n (%) *P. aeruginosa* *A. baumannii* *S. maltophilia*	36 (56.2) 12 (18.8) 16 (25.0)	21 (37.5) 22 (39.3) 13 (23.2)	0.035
Appropriate treatment, n (%)	26 (40.6)	24 (42.9)	0.854
Overall in-hospital mortality, n (%)	16 (25.0)	52 (92.9)	<0.001
14-day mortality, n (%)	0 (0.0)	43 (76.8)	<0.001
28-day mortality, n (%)	5 (7.8)	50 (89.3)	<0.001
Duration of antibiotic therapy/bacteremia, days, median (IQR)	15 (10–18.75)	4 (2–12)	<0.001
Length of hospital stay, days, median (IQR)	54 (32–86.75)	33 (16–50.5)	<0.001
Incidence of *C. difficile* infection, n (%).	2 (3.1)	3 (5.4)	0.663
90-day re-admission, n (%)	16 (25.0)	3 (5.4)	0.005

SD: standard deviation, IQR: interquartile range; MDRO: multidrug-resistant microorganisms; ICU: Intensive care unit.

**Table 5 microorganisms-11-00899-t005:** Multivariate logistic regression analysis of risk factors for clinical failure.

	Odds Ratio	95%CI	*p*-Value
Age	0.952	0.919–0.987	0.007
ICU admission	9.61	2.89–31.95	<0.001
Sepsis/Shock septic	5.19	1.80–14.96	0.002
Antibiotic therapy in previous 30 days	6.04	1.92–19.02	0.002
Contact with Healthcare facility	3.157	1.12–8.88	0.029

## Data Availability

Not applicable.

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
