# Peer review of "The Challenge of Bacteremia Treatment due to Non-Fermenting Gram-Negative Bacteria"

_microorganisms, 2023, doi:10.3390/microorganisms11040899_

Round 1

Reviewer 1 Report

Thank you for the opportunity to review the manuscript “The challenge of bacteremia treatment due to non-fermenting gram-negative bacteria”. This study provides a nice summary of the challenge in the management of multidrug-resistant Gram negative bacterial infection in the hospital setting. The overall methods were sound. Although I believe there are a few points that could be further improved.

Some parts of the methods were missing. First, the number of patients included in this study was a bit odd. The study duration was 6 years in a 1000-bed hospital, yet only 120 patients were included. Could the authors describe the inclusion and exclusion in more detail? Second, the production of carbapenemases was reported in the results, but the detail on how the authors detected the enzymes was missing. Third, a few definitions were also missing from the methods, such as multidrug resistance definitions for A. baumannii and S. maltophilia and the definition of “previous MDRO colonization”.

As the study period was long, it would be interesting to report the temporal change in the prevalence of resistance. Although I understand that this may be difficult given the rather low number of cases. There was a negative finding that I think was very interesting in this study. The author reported that the outcome was not associated with the appropriateness of the empirical treatment (p = 0.854). This is in contrast with popular belief. Could the author discuss more on this?

Lines 276 – 290: This paragraph described how monotherapy was associated with higher clinical success and why clinicians should avoid prescribing combination therapy. However, this was contradicted by the multivariate analysis in this study, which suggested that the use of monotherapy was not an independent factor. The author cited a study about S. maltophilia infection comparing TMP/SMX and minocycline treatment. The authors classified TMP/SMX as a combination therapy, which is unusual as the two agents share a similar target. Combination therapy usually refers to the use of two or more active agents that have different targets (like beta-lactams and aminoglycosides combination).

Finally, the English in this manuscript was difficult to understand. For example, in Lines 115 – 116, it should be something along the line of “the treatment was considered to be adequate if the organism was susceptible to the antimicrobial agent in vitro”. There was also inconsistency in capitalizing the drug name (see line 190 ‘tigecycline’ and line 193 ‘Tigecycline’).

Author Response

Dear Editor-in-chef,

We are grateful to the reviewers for their time and constructive feedback on our manuscript. We have implemented their comments and suggestions and wish to submit a revised version of the manuscript for further consideration of inclusion in your journal. Changes in the initial version of the manuscript are highlighted for added sentences. We are submitting two versions of the revised manuscript; one version with highlighted changes, and a clean version with all changes accepted.

Below, we also provide a point-by-point response explaining how we have addressed each of the reviewers’ comments. We look forward to the outcome of your assessment.

Yours sincerely,

On behalf of the co-authors

Svetlana Sadyrbaeva PharmD, PhD.

Response to Reviewer 1 Comments

Point 1: First, the number of patients included in this study was a bit odd. The study duration was 6 years in a 1000-bed hospital, yet only 120 patients were included. Could the authors describe the inclusion and exclusion in more detail? 

Response 1: Thank you for your comment.

Indeed, the number of bacteremia cases was considerably higher. The list we received from the microbiology laboratory had a total of 734 bacteremias caused by non-fermenting gram-negative bacilli, of which 528 were primary cases. Patients under 18 years of age were excluded.

We decided to analyze only the most frequent microorganisms causing nosocomial infections, such as Acinetobacter baumanii, Stenotrofomonas maltophilia and Pseudomonas aeruginosa. Isolates without sensitivity data were excluded. Finally, we only included the first episode for each patient.

In addition, during the years 2016-2018, there was a restructuring of the two hospitals in Granada (they were merged and then separated); therefore, for some patients during this period it was difficult to access their medical records. Patients belonging to the other hospital were thus excluded.

We have changed the order of some of the paragraphs to make it clearer.

Point 2: Second, the production of carbapenemases was reported in the results, but the detail on how the authors detected the enzymes was missing.

Response 2: We thank the reviewer for pointing this out. The detail of carbapenemase detection was included in the sub-section of Antibiotic susceptibility (lines 155-156).

Point 3: A few definitions were also missing from the methods, such as multidrug resistance definitions for A. baumannii and S. maltophilia and the definition of “previous MDRO colonization”.

Response 3: We thank the reviewer for pointing this out.

The definition of previous MDRO colonization were included in the sub-section of Variables and definitions (lines 134-136).

Furthermore, it was specified multidrug resistant definitions for A.baumannii and Stenotrophomonas (line 107-112).

Point 4: As the study period was long, it would be interesting to report the temporal change in the prevalence of resistance. Although I understand that this may be difficult given the rather low number of cases. There was a negative finding that I think was very interesting in this study. The author reported that the outcome was not associated with the appropriateness of the empirical treatment (p = 0.854). This is in contrast with popular belief. Could the author discuss more on this?

Response 4: Thank you for raising this point.

That's a very interesting suggestion; we will consider it in the future. Regarding the absence of an association between appropriate treatment and clinical success, this is probably due to the size of the sample. We include this observation in the text of the paper (lines 296-298).

Point 5: Lines 276 – 290: This paragraph described how monotherapy was associated with higher clinical success and why clinicians should avoid prescribing combination therapy.

Response 5.1:

It is not exactly like that. Moreover, in our study, patients with clinical success had antibiotic treatment in monotherapy against Gram-negative microorganisms. We explain because of the lower severity condition, as patients with clinical success had a lower rate of admission to the ICU, less Charlson index and Pitt score, among others.

However, this was contradicted by the multivariate analysis in this study, which suggested that the use of monotherapy was not an independent factor.

Response 5.2: Indeed, in the multivariate model, monotherapy was not found as a protective factor against clinical failure. We only referred to the bivariate analysis, explaining that the clinical success cohort probably consisted of less severely ill patients.

The author cited a study about S. maltophilia infection comparing TMP/SMX and minocycline treatment. The authors classified TMP/SMX as a combination therapy, which is unusual as the two agents share a similar target. Combination therapy usually refers to the use of two or more active agents that have different targets (like beta-lactams and aminoglycosides combination).

Response 5.3: It is probably not clearly explained in the paper, but what we mean is that TMP/SMX is usually used in monotherapy against Stenotrophomonas. However,it may not be a safe alternative as suggested by the published study, where the results were better with minocycline. Therefore, we suggest that at least at the beginning of the treatment a combined treatment should be considered (line 326-329)

Point 6: Finally, the English in this manuscript was difficult to understand. For example, in Lines 115 – 116, it should be something along the line of “the treatment was considered to be adequate if the organism was susceptible to the antimicrobial agent in vitro”. There was also inconsistency in capitalizing the drug name (see line 190 ‘tigecycline’ and line 193 ‘Tigecycline’).

Response 6: Thank you for this comment. We have fixed Tigecycline and modified the definition of appropriate treatment (lines 140-141)

Reviewer 2 Report

my comments

1- Add a reference from line 34 to line 42.

2- Change gram negative bacteria to Gram negative bacteria throughout the manuscript.

3- The material and method section should be divided into subtitles, such as patient sample, bacterial isolates identification, antibiotic susceptibility, statistical analysis, and add an appropriate references for each method.

4- In line 172, change microbiological isolations to bacterial isolates

5- In line 187 to 201 rephrase it to be more clear

6- You mentioned number (number) in all tables, for example 45(65). You should explain what you mean.

7- All abbreviations in all tables should be defined in the tables themselves.

8- Remove the paragraph from lines 310 to 317 because it is in the conclusion section.

Author Response

Dear Editor-in-chef,

We are grateful to the reviewers for their time and constructive feedback on our manuscript. We have implemented their comments and suggestions and wish to submit a revised version of the manuscript for further consideration of inclusion in your journal. Changes in the initial version of the manuscript are highlighted for added sentences. We are submitting two versions of the revised manuscript; one version with highlighted changes, and a clean version with all changes accepted.

Below, we also provide a point-by-point response explaining how we have addressed each of the reviewers’ comments. We look forward to the outcome of your assessment.

Yours sincerely,

On behalf of the co-authors

Svetlana Sadyrbaeva PharmD, PhD.

Response to Reviewer 2 Comments

Point 1: Add a reference from line 34 to line 42.

Response 1: added

Point 2: Change gram negative bacteria to Gram negative bacteria throughout the manuscript.

Response 2: changed

Point 3: The material and method section should be divided into subtitles, such as patient sample, bacterial isolates identification, antibiotic susceptibility, statistical analysis, and add an appropriate references for each method.

Response 3: division into subtitles performed and references added where applicable.

Point 4: In line 172, change microbiological isolations to bacterial isolates

Response 4: changed

Point 5: In line 187 to 201 rephrase it to be more clear

Response 5:

For some reason, the line count does not match the one I have. We guess the reviewer 2 refers to the antibiotic treatment subsection. The paragraph has also been rephrased form line 210 to line 216.

Point 6: You mentioned number (number) in all tables, for example 45(65). You should explain what you mean.

Response 6: Indeed, as indicated in column 1 (Variables, n (%)). Next to each variable appears n(%) or IQR, therefore 45 is the “n” and (65) is the percentage (65%)

Point 7: All abbreviations in all tables should be defined in the tables themselves.

Response 7: done.

 Point 8: Remove the paragraph from lines 310 to 317 because it is in the conclusion section.

Response 8: done.

Reviewer 3 Report

Please review and comment on the discussion:

The burden of bacterial resistance:

Lancet Public Health 2022; 7:e897.

Difficult to treat resistance and standard definitions for acquired resistance:

Magiorakos AP. Clin Microbiol Infect 2012; 18:268.

Kadri SS. Clin infect Dis 2018; 67:1803.

Huh K. Clin Infect Dis 2020; 71:e487.

Time to positivity in blood cultures and follow-up blood cultures:

Hsieh YC. BMC Infect Dis 2022; 22:142.

Mitaka H. Infect Control Hosp Epidemiol 2022; 29:1.

Other laboratory tests:

Lien F. BMC Infect Dis 2022; 22:287.

Banerjee R.  Clin Infect Dis 2021; 73:e39.

Cumulative antibiograms and decision-making in empiric antibiotic selection:

Teitelbaum D. Clin Infect Dis 2022; 75:1763.

Elligsen M. Clin Infect Dis 2021; 73:e417.

Ohnuma T. Jama Network Open 2023; 6;e2249353.

And:

Surviving sepsis and during covid-19:

Evans L. Intensive Care Med 2021; 47:1181.

Afzal A. Int J Infect Dis 2022; 116:43.

Author Response

Dear Editor-in Chef

We are grateful to the reviewers for their time and constructive comments on our manuscript. We have implemented their comments and suggestions and wish to submit a revised version of the manuscript for the further consideration in the journal. Changes in the initial version of the manuscript are either highlighted for added sentences. We are submitting a two version of the revised manuscript; a version with the changes highlighted, and a clean version with all changes accepted.

Below, we also provide a point-by-point response explaining how we have addressed each of the reviewers’ comments. We look forward to the outcome of your assessment

Yours sincerely,

On behalf of the co-authors

Svetlana Sadyrbaeva PharmD,PhD.

Response to Reviewer 3 Comments

Please review and comment on the discussion:

The burden of bacterial resistance:

 Lancet Public Health 2022; 7:e897. cited

 Difficult to treat resistance and standard definitions for acquired resistance:

 Magiorakos AP. Clin Microbiol Infect 2012; 18:268. cited

 Kadri SS. Clin infect Dis 2018; 67:1803.cited

 Huh K. Clin Infect Dis 2020; 71:e487. cited

 Time to positivity in blood cultures and follow-up blood cultures:

 Hsieh YC. BMC Infect Dis 2022; 22:142.

 Mitaka H. Infect Control Hosp Epidemiol 2022; 29:1.

 Other laboratory tests:

 Lien F. BMC Infect Dis 2022; 22:287.

 Banerjee R.  Clin Infect Dis 2021; 73:e39.

 Cumulative antibiograms and decision-making in empiric antibiotic selection:

 Teitelbaum D. Clin Infect Dis 2022; 75:1763.

 Elligsen M. Clin Infect Dis 2021; 73:e417.

 Ohnuma T. Jama Network Open 2023; 6;e2249353. cited

 And:

 Surviving sepsis and during covid-19:

 Evans L. Intensive Care Med 2021; 47:1181.

 Afzal A. Int J Infect Dis 2022; 116:43.

Response 1: Thank you very much for suggesting these articles, we have found them very interesting and helpful to improve the article. However, we have not cited all of them because some of them are not related to the objective of the article. We underline the ones we have cited. We highlight the articles that we have cited.